# Does the Invasive *Heracleum mantegazzianum* Influence Other Species by Allelopathy?

**DOI:** 10.3390/plants13101333

**Published:** 2024-05-12

**Authors:** Daniela Gruľová, Beata Baranová, Adriana Eliašová, Christelle Brun, Jozef Fejér, Ivan Kron, Luca Campone, Stefania Pagliari, Ľuboš Nastišin, Vincent Sedlák

**Affiliations:** 1Department of Ecology, Faculty of Humanities and Natural Sciences, University of Prešov, 17. Novembra 1, 080 01 Prešov, Slovakia; beata.baranova@unipo.sk (B.B.); adriana.eliasova@unipo.sk (A.E.); brun_christelle@hotmail.com (C.B.); jozef.fejer@unipo.sk (J.F.); kron.ivan@gmail.com (I.K.); lubos.nastisin@nppc.sk (Ľ.N.); 2Department of Biotechnology and Bioscience, University of Milano-Bicocca, 20126 Milan, Italy; luca.campone@unimib.it (L.C.); stefania.pagliari@unimib.it (S.P.); 3Research and Breeding Station Malý Šariš, Research Institute of Plant Production Piešťany, National Agricultural and Food Centre Nitra, Malý Šariš 221, 080 01 Prešov, Slovakia; 4Department of Biology, Faculty of Humanities and Natural Sciences, University of Prešov, 17. Novembra 1, 080 01 Prešov, Slovakia; vincent.sedlak@unipo.sk

**Keywords:** invasive plant, phytotoxic activity, antioxidant, phenolics, coumarins, LC-MS furocoumarins, seed germination

## Abstract

*Heracleum mantegazzianum* is an invasive species in middle Europe. The mode of action of its invasiveness is still not known. Our study focuses on observation of potential allelopathic influence by the production and release of phytochemicals into its environment. Plant material was collected four times within one season (April, May, June, July 2019) at locality Lekárovce (eastern Slovakia) for comparison of differences in composition and potential allelopathy. Water extracts from collected samples were used for different biological assays. The total phenols and flavonoids were determined spectrophotometrically. The profile and content of phenolic components, including coumarins, were determined by two techniques of liquid chromatography along with in vitro evaluation of the free radical scavenging activity of extracts (DPPH, Hydroxyl, Superoxide, and FRAP). The changes in composition in extracts in different seasonal periods were evident as well as potential phytotoxic activity in some concentrations on specific model plants. The slight antioxidant activity was noted. The invasiveness of the current species could be supported by the excretion of its phytochemicals into its surroundings and by different modes of action influencing living organisms in its environment.

## 1. Introduction

One of the most studied families among the plants is Apiaceae. It contains 300 genera and about 3000 species. Their secretion ducts contain a great number of secondary metabolites [1,2,3,4,5,6]. Phytochemicals could have beneficial or harmful effects on other organisms in their germination, growth, or development of different organs [7,8]. The secondary metabolites, such as phenolics, flavonoids, alkaloids, terpenoids and cyanogenic glycosides, have often attracted scientists to elucidate their structure and biological function [9,10]. Most of the allelopathic compounds released are hydrophilic, such as phenolic acids, alkaloids, flavonoid glycosides, etc. [9,11,12,13,14,15,16,17]. Plants produce phytochemicals which evaporate from aboveground parts such as flowers or leaves, as well as from underground plant parts such as roots, to their surroundings [1].

The genus *Heracleum* belongs to the Apiaceae family and is native to many regions of the world [18]. *H. mantegazzianum* is the only species from the genus *Heracleum* identified in Slovakia as an invasive plant [19]. The plant has huge morphology, reaching up to 3–5 m in height. Based on monitoring, it spreads by seeds, usually along rivers [20]. This species occurs on the territory of Poland, Czech rep., Slovakia and Hungary. Due to its size, high reproduction, early germination and growth, it is a suitable type of invasive plant [6,21]. As a result of increasing knowledge of *H. mantegazzianum* as an invasive species with a toxic effect, its introduction also could be accidental, such as by transfer on car tires or the collection of dry flower heads for decorative purposes [22,23,24,25]. There are some indications that they suppress original flora [26]. Discussions by researchers on what allows them to spread remain inconclusive. One of the hypotheses is based on their secondary metabolites, which leach into the soil and potentially reduce the viability and germination of seeds of other species [27]. *H. mantegazzianum* produces different phytochemical groups. Among those identified were essential oils, which can be hydro distillated from the seeds in large amounts [28,29,30]. In its water, methanolic, ethanolic, or other kinds of extracts the phenols, flavonoids and furanocoumarins were identified [1,6,17,31]. Many studies describe the biological activities of essential oils or extracts of various *Heracleum* species as antimicrobial, antifungal, antioxidant, phytotoxic, immunostimulant, etc. [6,25,30,31,32,33]. Based on the research premise, the allelopathic effect is species-specific, and the secretion of the active components can be highly seasonal and can even vary substantially between years [34,35]. Furanocoumarins, which are present in *Heracleum mantegazzianum*, could contribute to its allelopathic potential and spread to new locations [27,36]. The publication of *Heracleum mantegazzianum* mostly focuses on its essential oil. The composition and the biological activity of other phytochemical groups of the mentioned species have not been precisely investigated.

Our investigation was established on the simulation of natural conditions, where the compounds are released into the soil by the plant litter or by the influence of other, exogenous factors. When the chemicals are already in the soil, the hypothesis is that they can have a direct impact on the seeds of other plant species. Additional to the hypothesis was that if the allelopathic activity of *Heracleum mantegazzianum* was found to be effective, that finding could be used in the development of a potentially more ecological variant of plant-based herbicide.

The aims of the study were (1) to evaluate the allelopathic impact of *Heracleum mantegazzianum* extracts from different seasons or periods on four types of model plant seeds—*Raphanus sativus* L. (radish), *Sinapis alba* L. (white mustard), *Triticum aestivum* L. (summer wheat) and *Hordeum vulgare* L. (barley); (2) to compare the composition of the extracts during the vegetative phase by determining the content of phenols, flavonoids and coumarins; and (3) to evaluate the potential antioxidant activity using different methods.

## 2. Results

### 2.1. Total Phenols, Total Flavonoids and Free Radical Scavenging Activity of the Leaves and Seed Extracts

The extract from the leaves of *Heracleum mantegazzianum* showed a statistically significant higher amount of dry matter and total phenolic and total flavonoid content compared to the seed extract (Table 1). A statistically significant higher antioxidant activity against the superoxide radical was also found in the leaf extract compared to seed extract. Correlation analysis confirmed a high positive correlation of r = 0.945 with the total content of phenolic substances. Similarly, the leaf extract showed significantly stronger DPPH scavenging ability than the extract prepared from seeds (Table 1). In the same assay, the potent antioxidant compound ascorbic acid reached an IC_50_ value of 2.72 (±0.031 µg·mL^−1^), representing approximately 60- and 140-fold greater anti-free-radical effect compared to the leaf and seed extracts, respectively.

Opposite trends were found in the antioxidant activity against the hydroxyl radical and in the FRAP method. Seed extracts showed statistically significantly higher antioxidant activity compared to leaf extracts. Correlation analysis showed a high negative correlation coefficient in relation to the total phenols content, r = −0.961 and r = −0.973, respectively.

All of the methods were used to confirm the total antioxidant activity of the extracts, and each confirmed the effect of a different spectrum of phenolic compounds.

### 2.2. Phenolic Profile of Extract Obtained from April to July

The composition of the extracts was analyzed by HPLC-DAD, and the putative identities of 22 chromatographic peaks were determined using available standard substances (Table 2). Most of the phenolic components, 15 in number, were the mono- and di-O-glycosides of the flavone quercetin and kaempferol, respectively. Neither quercetin-3-O-rutinoside (rutin) nor quercetin-3-O-galactoside (hyperoside) were detected in the extracts. The group of hydroxycinnamic acids was represented by neochlorogenic acid, chlorogenic acid, caffeic acid, and p-coumaric acid, including some caffeic acid derivative, as indicated by the nature of its UV-Vis absorption spectrum. The remaining two components of the extracts were protocatechuic acid and an unidentified coumarin derivative (distinct from umbelliferone). These substances were present in each of the four extracts compared. However, there were significant differences in the content of individual substances between the extracts, along with significant differences in the content of total phenols and total flavonoids (results of ANOVA, *p* < 0.001 for each parameter evaluated). The di-glycosides of kaempferol and quercetin, respectively, as well as chlorogenic acid were the dominant constituents of the extracts prepared from *Heracleum mantegazzianum* leaves collected in April and May. In the case of the latter, caffeic acid, quercetin-3-O-glucoside and kaempferol-3-O-glucoside were also detected in large amounts, unlike all other extracts. In the extracts prepared from plants collected in June and July, respectively, a putative coumarin derivative was observed as the predominant constituent. This substance was also present in a comparable quantity in the May extract, while its content in the April extract reached less than half the value. Overall, the extract from plant material collected in May was characterized by the highest content of total phenols and total flavonoids. The results obtained indicate that changes in the production of individual phenolic compounds in *Heracleum mantegazzianum* leaves occur during the season.

Considering the presence of several molecules belonging to the coumarin and furanocoumarin classes in the Apiaceae family, these were characterized by UHPLC-HRMS/MS analysis to identify them and assess their qualitative and quantitative variation during the vegetative phase. The chromatographic conditions of the UPLC method were optimized to achieve efficient separation of the analytes, high peak resolution, and efficient ionization. Mass experiments were performed in positive ionization modes. Metabolites were tentatively identified using UV spectra, HRMS data (accurate mass, isotopic pattern, molecular formula) and MS/MS fragmentation pathway, and compared with literature databases. UPLC-ESI-Q/TOF-MS analysis identified eight coumarins and furocoumarins. They are secondary metabolites from polyphenols belonging to the benzopyrionic compounds, consisting of benzene rings fused to α-pyrone rings found in many plant families, including Apiaceae. They have been identified based on their specific fragmentation path in the positive ion current. Indeed, coumarins show a characteristic fragmentation with loss of -CO (28 Da) and -CO_2_ (44 Da) groups from the protonated molecular formula [M+H]+. Furthermore, the loss of the methyl group -CH_3_ (15 Da) can be observed in the case of methoxylated coumarins and the loss of H_2_O (18 Da) in all hydroxylated coumarins [37,38,39]. Based on this information and the use of the standard, compound **8** was identified as isopimpinellin. In the MS spectrum obtained, the molecular ion *m/z* 247.0611 was assigned the molecular formula C_13_H_11_O_5_, which was subsequently identified as isopimpinellin thanks to the fragmentation pattern obtained because of the tandem MS analysis. The product ions *m/z* 232.0377 and *m/z* 217.0143 were generated by the loss of one methyl group (15 Da) and two methyl groups (30 Da), respectively, from the precursor ion. On this basis, compounds **1**, **2**, **3**, **4**, **5**, **6**, **7** and **8** were probably identified (Table 3).

Semi-quantitative analysis (Table 4) was performed to identify differences in the distribution of coumarins in different seasons (spring and summer). Compounds for which authentic standards were not available were quantified to standard equivalents using the closest chemically related standard available. The results showed a higher presence of simple coumarins such as hydroxycoumarin isomers 1 (22.892 ± 1.174 (April) and 23.238 ± 1.025 (May) mg·g^−1^ DM) in spring compared to summer, whereas in June and July, there was a higher presence of some furanocoumarins such as angelicin (11. 930 ± 0.220 (June) and 9.990 ± 0.307 (July) mg·g^−1^ DM) and psoralen (3.768 ± 0.152 (June) and 3.737 ± 0.100 (July) mg·g^−1^ DM) compared to spring.

### 2.3. Phytotoxic Activity

When comparing to control, the pure extract from *Heracleum mantegazzianum* leaves collected in April, May and July caused significant declines in the percentages of *Raphanus sativus* and *Sinapis alba* germinated seeds (Table 5). Consequently, significant differences were observed also in comparison to aqueous solutions of extracts with lower content of dry mass. No germination at all was observed when using pure extract from June leaves which had the highest content of dry mass. Significant anti-germinative activity against *R. sativa* and *S. alba* was noticed also by application of the May and June extracts in doses of 50 mL and 100 mL, and also against *S. alba* by application of the April and May extracts in doses of 25 mL and 100 mL, respectively. Other extracts with lower dry mass content did not affect seeds germination at all. A significant negative linear dependence was observed between *S. alba* and *R. sativa* germination and the content of dry mass. In contrast, *Triticum aestivum* and *Hordeum vulgare* showed higher germination rates (Table 5). A significant decline in the seed germination rate was, in comparison to control, observed in *H. vulgare* after application of the April and June pure extracts, and in *T. aestivum* only after application of the July pure extract.

Concerning root growth (Table 6), with the only exception being *H. vulgare* after application of the July pure extract, roots of *T. aestivum* and *H. vulgare* were significantly shorter in comparison to control after application of pure *Heracleum mantegazzianum* extract from leaves collected in April, May, and June as well as July. The same concerned *R. sativa* and *S. alba.* Based on the obtained results, monocot species seemed to be less susceptible to the anti-germinative activity of *Heracleum mantegazzianum* leaf water extracts when compared to those from the dicot group. According to linear regression, the germination rate as well as root growth significantly decreased as the concentration of applied extract increased, and this was observed for all model organisms.

Although there were significant differences between the April, May, June and July extracts in dry mass content, total phenols content, total flavonoids content, content of individual phenolic acids, flavonoids, and coumarins, no significant relationship between those quantitative parameters and the seed germination rate or seedling root growth was detected with our set of data. Then, mutual comparison of the April, May, June and July pure extracts’ phytotoxic activity showed the following results: no differences were observed in the percentages of germinated seeds or root length of *R. sativa* and *S. alba*. However, germination as well as root growth were in general very low or even zero. Different results were obtained with monocot species: the lowest percentages of germinated seeds as well as root length of *T. aestivum* and *H. vulgare* were connected with the pure extracts from leaves collected in April, in predominant cases even significantly lower in comparison to May, June and July.

## 3. Discussion

The composition of phytochemicals in different plant species of the Apiaceae family was determined many years ago [2,43], as well as the allelopathic effect of phytochemicals of one species on another [44,45]. The exact effects of specific species are still being discovered. The current article presents a study of water extracts of *Heracleum mantegazzianum* on selected model plants. In this study, the dominant components were identified, and total phenolic, flavonoid and coumarin composition was determined. For comparison, in another study of the phytochemical and biological activity of nine plant species belonging to the Apiaceae family, one species was *Heracleum persicum*. Total phenolic (TPC) and flavonoid (TFC) content was determined in the methanol extract using methods similar to those in our study. The free radical scavenging activity was also measured by the same methods using DPPH [37]. TPC was determined to be 9.58 ± 0.47 mg GAE/g DW, which was about one-third to half the value obtained with our measurements. TFC was determined to be 2.82 ± 0.18 mg QE/g DW, which was very similar to the results of our seed extract determination. DPPH IC_50_ was noted at 15.82 ± 0.93 mg/mL, which was also comparable with our results measured in leaf extracts. The high TPC and TFC as well as antioxidant activity (DPPH) in different plant parts were confirmed by a study on *Heracleum platytaenium* methanolic extract [38]. The changes in the chemical profile during the vegetative phase in the aboveground plant parts of *H. persicum* were determined in the study [34]. The highest phenolic acid contents were determined in the mid-season (in the floral bud stage in June). Compared to our analysis, the highest amount of TPC was noted in the beginning of mid-season (in May). Researchers noted that changes in the chemical profile strongly depend on the plant phenology stage. Plant development in the season also depended on climatic conditions such as air temperature and humidity [35], which influenced the speed of plant maturation in the year of the investigation. In different plant families, chemical compounds called coumarins were identified. They are very typical for the family Apiaceae, especially of the tribe to which the genus *Heracleum* belongs. Apiaceae family members are more famous for their sub-group furanocoumarins [39]. In the *Heracleum* spp. extracts, coumarins and furocoumarins are naturally present in roots, fruits and leachates [40]. Furanocoumarins exert their toxicity in different ways. Furthermore, furanocoumarins can inhibit enzymes as well as bind to proteins and to unsaturated fatty acids [41,42]. A combination of several furanocoumarins usually exhibits higher toxicity than each compound alone [42,46]. Phytotoxic effects of plant coumarins were identified and presented as those of a potential new generation of bioherbicides [47]. In our study, we identified components belonging to the coumarin group similar to those identified by other researchers [1]. Their seasonal changes were also confirmed. *H. mantegazzianum* contains the second highest concentration of furanocoumarins after *Ammi visnaga* L., and their differences in concentrations were noted in different plant parts as well as within the seasons [48,49,50,51]. The concentration of coumarins and furanocoumarins in the plant depends on various factors such as the maturity of the plant, harvesting conditions and climatic conditions, particularly temperature, UV exposure and rainfall [52]. Researchers identified linear as well as angular furanocoumarins in *Heracleum mantegazzianum*. The dominant furocoumarins were identified as bergapten, xanthotoxin, imperatorin, isopimpinellin, psoralen angelicin pimpinellin, and in the roots, sphondin and isobergapten [6,31,39,48,53,54]. Similar components were identified in *H. leskowskii* [17] and *Heracleum candicans* [55] as well as *H. persicum* [56]. Indeed, studies on *Heracleum* spp. and other plant species have found that the production of furanocoumarins such as psoralen, bergaptene and angelicin can increase after exposure to high temperatures, probably due to increased ROS production in cells [52,57,58]. A report on the possible allelochemical influence of invasive species is discussed in Ref. [43]. The report mentions the increasing concentrations of furanocoumarins in the new localities for specific plant species and, based on this, their impact on other species. Investigations of plant chemical composition also yielded knowledge that many of them are directly or indirectly involved in plant defense [26,42,59,60]. The potential inhibitory activity of *Heracleum sosnowskyi* toward other plant species (*Alium cepa*) was investigated in recent studies [61,62]. Aqueous solutions of leaf extract were used to test the biological activity, and root length was reduced. A depressed mitotic effect on meristematic cells of *A. cepa* roots was presented as an explanation. Compared to our study, which tested the direct impact of plant extracts on the seeds of model plants, an indirect study was provided by Ref. [7]. Researchers used soil from locations invaded by *H. mantegazzianum* in which expected secondary metabolites leached from the mentioned species. Three plant species were placed into this soil for cultivation (*H. mantegazzianum*, *Rumex obtusifolius* L. and *Urtica dioica* L.). Reduced germination of *U. dioica* seeds on the invaded soil was noted as a positive result. They also used aqueous extract from this soil to evaluate the germination of *Lapsana communis* L. and *R. obtusifolius*. No significant effect was noted. Then, the next experiment was aimed at determining the allelopathic effects of moist seeds of *H. mantegazzianum* on the germination of nine plant species (*Brachypodium sylvaticum* (Huds.) P.B., *Calystegia sepium*, *Euphorbia helioscopia* L., *Festuca gigantea* L., *Mentha arvensis* L., *Poa trivialis* L., *R. obtusifolius* L., *Vicia hirsuta* (L.) Grey and *U. dioica*). *C. sepium* showed reduced germination in this variation of the experiment. The fourth experiment tested the effects of *H. mantegazzianum* seed extracts on the seed germination of *M. arvensis*, *P. trivialis*, *Sonchus oleraceus* L. and *U. dioica*. The seed extract negatively affected the germination of *P. trivialis* and *U. dioica* seeds [7]. In natural conditions, there are several factors which can influence seed germination, such as soil/air temperature, moisture, soil microbiota and unknown concentrations of allelochemicals. In the present research, the direct impact of the chemical compounds was more noticeable. Furthermore, changes in soil pH or nutrient concentrations in stands invaded by *H. mantegazzianum* could explain differences in the seed germination of other species [7]. Another researcher [63] found inhibitory effects of the furanocoumarins in *Heracleum laciniatum* on the germination of lettuce, as also supported by [64]. A recent study on coumarins as allelopathic agents comes from [47]. In addition to the above, phenolic compounds determined in the invasive species *Merremia umbellata* subs. *orientalis* showed inhibition of *Arabidopsis* seed germination. Through this experiment, researchers supported the role of phenolic compounds as allelochemicals responsible for the success of plant invasiveness [65]. From the above-mentioned information, it is clear that many investigations have been conducted on different species from the genus *Heracleum*. In Slovakia, *H. mantegazzianum* is an invasive plant. Our research presented study following a previous investigation [30]. We have enriched study of the potential allelopathic influence of different phytochemical groups extracted from the mentioned species. Based on our hypothesis, the potential effects on other plant species were evaluated, but with specific doses in specific target model plants. Generalization of the findings of the current evaluation is not possible. The current research can be a starting point for deeper study aimed at understanding the mode of action at different levels. Additional experiments are needed to understand the metabolism of phytochemicals and their influence on primary metabolism in plants. In the presented study, we can conclude that living organisms have an influence on each other, but the main factors responsible for blocking seed gemination or root elongation are still open to question. From another perspective, it is necessary to consider many other factors, such as the genetics of the studied plant species, climatic conditions in the collection area, and the preciseness of the application of different methods. Nevertheless, our question as posed in the title of this research article was answered: *H. mantegazzianum* does have allelopathic effects on other species.

## 4. Materials and Methods


**Plant material**


*Heracleum mantegazzianum* is the only species from genus *Heracleum* identified in Slovakia [19]. Fresh plant material was collected four times in the vegetative season of 2019 from grassland near the river Uh. Each collection was performed on a dry sunny day once in April, May, June, and July at the locality of Lekárovce (48°36′12.2962137″ N, 22°9′30.3183174″ E) in southeastern Slovakia. After collection, each sample (4 in total) was transferred to the laboratory at the University of Prešov, where the plant leaves were spread on filter paper and left until dry at room temperature. After that, dry plant material was pulverized into powder in an electrical mill (ORAVA, RM 1550, Bratislava, Slovakia) and stored in a dry place until analysis.


**Water extract preparation**


Twenty grams of plant powder from one sample was mixed with 200 mL of distilled water in an Erlenmeyer (EM) bank. The prepared solution was placed in a water bath (60 °C) for 1 h. A cooler was placed on top of the EM bank to avoid water evaporation. Extract was prepared in three replications. When the extract was cooled, it was filtrated by using a Büchner funnel connected to a vacuum pump (V-700, Vacuum Pump, Büchi, Flawil, Switzerland). Filtration was repeated two times. After filtration, extracts were also centrifugated at 6000 rpm for 30 min (Hettich Zentrifugen, Universal 320). Water extract was placed in a freezer until analysis.


**Dry mass evaluation**


The dry mass of each prepared extract was evaluated in four repetitions. One milliliter of extract was placed into a glass Petri dish and placed into an oven at 105 °C until reaching constant weight.


**Determination of Total Phenolic Content**


Total content of phenolic compounds was determined with the Folin–Ciocalteu reagent (FCR, Merck, Rahway, NJ, USA) according to the published procedure [66], with slight modifications [67]. Briefly, 0.1 mL of appropriately diluted extract, 0.2 mL of FCR, 2 mL of double distilled water (DDW), and 1 mL of Na_2_CO_3_ water solution (20%, *w/v*) were well-mixed in a test tube. The reaction mixture was kept in the dark at room temperature for 90 min. The absorbance of the mixture at λ = 765 nm was measured with a Shimadzu UV-1800 spectrophotometer using a blank in which the extract was replaced by DDW. The amount of polyphenols in the extracts was calculated using the calibration line of gallic acid (Merck) and expressed as gallic acid equivalents (GAE) mg·g^−1^ dry weight. The determinations were performed at least four times for each extract. All solutions were used on the day of preparation.


**Determination of Total Flavonoid Content**


Total content of flavonoids was determined by the aluminum chloride colorimetric method according to the published procedure [68], with slight modifications [67]. Briefly, 0.2 mL of appropriately diluted extract, 1.8 mL of DDW, 0.1 mL of AlCl_2_ water solution (10%, *w/v*), 0.1 mL of 1 M CH_3_COOK, and 2.8 mL of DDW were well-mixed in a test tube. The reaction mixture was kept at room temperature for 30 min. The absorbance of the mixture at λ = 415 nm was measured with a Shimadzu UV-1800 spectrophotometer using a corresponding blank in which the AlCl_2_ solution was replaced by DDW. The amount of flavonoids in the extracts was calculated using the calibration line of quercetin (Sigma Aldrich, St. Louis, MO, USA) and expressed as quercetin equivalents (QE) mg·g^−1^ dry weight. The determinations were performed at least four times for each extract. All solutions were used on the day of preparation.


**HPLC-DAD Secondary Metabolites Profiling Analysis**


The composition of the extracts was analyzed by the gradient reversed-phase high-performance liquid chromatography-diode array detector method on a Dionex UltiMate 3000 Quarternary Analytical LC System (Germering, Germany). Prior to analysis, the extracts were filtered through a syringe filter (0.45 μm, nylon membrane, Whatman Puradisc 13, Maidstone, UK). A Kromasil 100 C18 analytical column (5 μm, 250 × 4.6 mm; Nouryon, Sweden) kept at a temperature of 25 °C was used to separate the individual phenolics. The mobile phase consisted of solvent A (0.1% formic acid in HPLC gradient-grade water, *v/v*) and solvent B (0.1% formic acid in HPLC gradient-grade acetonitrile, *v/v*). The gradient elution program was as follows: 0 min, 2% B; 0–22 min, 2–20% B; 22–40 min, 20–40% B; 40–45 min, 40–100% B; 45–50 min, 100% B; 50–52 min, 100–2% B; 52–60 min, 2% B. The mobile phase flow rate was kept at 1.0 mL.min^−1^. The sample injection volume was 5 µL. The detection wavelength was set at 280, 320, and 350 nm. During the analyses, UV-Vis absorption spectra were recorded between 200 and 500 nm in 1 nm steps.

Identification of chromatographic peaks was based on their retention time and absorption spectra by comparison to the standard compounds. Commercially available standards caffeic acid, chlorogenic acid (Acros Organics, Geel, Belgium), neochlorogenic acid (Extrasynthese, Genay, France), protocatechuic acid, *p*-coumaric acid, kaempferol-3-O-glucoside, kaempferol-3-O-rhamnoside, quercetin-3-O-glucoside, quercetin-3-O-galactoside (Sigma Aldrich), quercetin-3-O-rhamnoside (Carl Roth, Karlsruhe, Germany), and quercetin-3-O-rutinoside (Acros Organics), together with substances isolated previously [69] whose identity was determined by the NMR method (kaempferol-3-O-glucosyl-7-O-rhamnoside and keampferol-3,7-di-O-rhamnoside) or by the HPLC-DAD-ESI-MS^2^ method (quercetin-6-deoxyhexose, -6-deoxyhexose and quercetin -6-deoxyheose, -hexose), were used. The content of individual phenolics in extracts was determined by the external standard calibration method and expressed as μmol·g^−1^ dry weight. All glycosides of quercetin and kaempferol were evaluated at λ = 350 nm using calibration lines of the standards quercetin 3-O-glucoside and kaempferol-3-O-glucoside, respectively. Chlorogenic acid and neochlorogenic acid were evaluated at λ = 320 nm using the calibration line of the chlorogenic acid standard. Caffeic acid and a putative derivative of caffeic acid were evaluated at λ = 320 nm using the calibration line of the caffeic acid standard. For the evaluation of *p*-coumaric acid (at λ = 320 nm) and protocatechuic acid (at λ = 280 nm), the calibration lines of the corresponding standards were used. A putative derivative of coumarin was evaluated at λ = 320 nm using the calibration line of the umbelliferone standard (7-hydroxycoumarin, Acros Organics).


**Characterization of *Heracleum mantegazzianum* extracts by UHPLC-ESI-HRMS analysis**


*Heracleum* spp. extracts were characterized using a Waters ACQUITY UPLC system coupled to a Waters Xevo G2-XS Q-TOF mass spectrometer (Waters Corp., Milford, MA, USA) using a method described by [70] with some modification to improve the separation and ionization of compounds. A Kinetex Biphenyl column (100 × 2.1 mm; 2.6 µm) was used for the analysis, and the mobile phases were prepared with MS grade H_2_O (A) and CH_3_CN (B) solvents, both containing 0.1% formic acid (HCOOH). The elution gradient was optimized as follows: 0–2.0 min, 5–10% B; 2.0–17.0 min, 10–35% B; 17.0–18.0 min, 35–95% B, to obtain a good separation of the analytes. After each run, 5 min of washing (98% B) and 5 min of conditioning were performed to restore the initial conditions. The column temperature was maintained at 30 °C throughout the analysis, and elution was performed at a flow rate of 400 µL.min^−1^ and an injection volume of 10 µL. UV spectra were acquired in the range 210–400 nm, with additional wavelengths set at 330 nm for better detection of the analytes of interest. Untargeted mass analysis was performed in positive ionization mode to obtain a full scan MS, and spectra were recorded in the *m/z* range 100–1000. Source parameters were optimized to achieve efficient ionization of the analytes as indicated: electrospray capillary voltage 2.5 kV, source temperature 150 °C and desolvation temperature 500 °C. The cone and desolvation gas flows were set to 10 and 1000 L·h^−1^, respectively. A scan time of 0.5 s was used. The cone voltage was set at 50 V, and the collision energy was defined with a ramp from 6 to 30 V to produce abundant product ions. The mass spectrometer was calibrated with 0.5 M sodium formate, and 100 pg/µL leucine-enkephalin at *m/z* 556.2766 (positive ionization) was used as a reference to achieve high mass accuracy. Lockmass was injected at 5 µL·min^−1^ simultaneously with the column flow and acquired for 1 s every 30 s. Baseline peak identification (BPI) chromatograms were acquired at low (6 V) and high (30 V) energy for peak identification. The molecular ion masses [M+H]+ were obtained from the low-energy spectra and used to determine elemental composition (mass error < 5 ppm), while the high-energy spectra provided the fragmentation pattern for identification. Compounds were characterized based on corresponding spectral features (UV and MS, [M+H]^+^), accurate mass, characteristic fragmentation and information from various databases (PubChem, Chemspider and KEGG). Stock solutions (1 mgmL^−1^) of coumarin, bergaptene and isopimpinelline were prepared for semiquantitative analysis. The calibration curve was constructed by HPLC-UV (wavelength at 330 nm), increasing the concentration of the solutions (0.1, 0.5, 1, 5, 10, 50 µgmL^−1^) by injecting 5 µL in technical triplicate. The calibration curve was obtained by linear regression using Excel 2016 software (Microsoft Corporation, Washington, DC, USA) considering the area of the external standard against the known concentration of each compound. The calibration curves for each standard showed good linearity with correlation coefficients (R^2^) ranging from 0.997 to 0.999. Mass Lynx software (version 4.2) was used for instrument control, data acquisition and data processing. Molecular identification was performed using UNIFI^®^ software v. 1.9, standards and literature data.


**Antioxidant activity in vitro tests**


Four different methods for antioxidant activity evaluation were performed. Plant material from leaves and seeds collected in July 2019 were used to prepare extracts by the method described above.


**
*2,2-Diphenyl-1-picrylhydrazyl (DPPH) Radical Scavenging Assay*
**


Free radical scavenging activity of the extracts was evaluated by DPPH radical scavenging assay [71]. The determination procedure was described in detail in our previous work [72]. Ascorbic acid (Sigma Aldrich) was used as reference antioxidant. Half maximal inhibitory concentration (IC_50_) value, i.e., the concentration of the plant extract/reference antioxidant that could scavenge 50% of DPPH radical value was calculated and expressed as µg mL^−1^. All determinations were performed at least four times for each extract/reference compound. All solutions were used on the day of preparation.


**
*Superoxide Anion Radical Scavenging Activity*
**


The experiment was prepared based on the published method [73] with slight modifications presented in previous publications [67,74,75]. Phosphate buffer (PB, sodium dihydrogenphosphate dihydrate and disodiumhydrogen phosphate dodecahydrate) in a concentration of 0.05 mol/L and pH 7.4 was used with 0.1 mmol/L Na_2_EDTA. Hypoxanthine (HX, Alfa Aesar a Johnson Matthey Company, Heysham, UK) 0.4 mmol/L was dissolved in the phosphate buffer. Then, 0.01 g of xanthine oxidase (XO, Sigma Aldrich, activity 0.11 units/mg solid or 0.713 units/mg proteins) was dissolved in 20 mL of phosphate buffer. Nitro blue tetrazolium chloride (NBT, Sigma Aldrich) 5 mmol/L in phosphate buffer was used as an indicator of superoxide radicals. Antioxidant activity of tested samples against superoxide radicals was compared with antioxidant activity of salicylic acid (SA) 10 mmol/L in the phosphate buffer. The solutions were pipetted into test tubes in duplicate. Solutions were mixed well and incubated in a water bath at 38 °C for 40 min. After incubation and cooling, the absorbance of the solutions was determined in a 1 cm cuvette at 560 nm, using the spectrophotometer Shimadzu UV-1800. The percentage of inhibition (Superoxide [%]) was calculated. All determinations were performed at least four times.


**
*Hydroxyl Radical Scavenging Activity*
**


The experiment for scavenging hydroxyl radicals based on deoxyribose was inspired by the published method [76] with small modifications [67,74,75]. Phosphate buffer (PBS, NaH_2_PO_4_/Na_2_HPO_4_) 0.05 mol/L pH 7.4 was used with 0.1 mol/L of NaCl and 9 mmol/L 2-deoxyribose. Then, 3 mmol/L ferrous sulfate heptahydrate was dissolved in 100 mL double-distilled water (DDW) with addition of 0.1 mL concentrated sulfuric acid to prevent oxidation of Fe(II) and hydrolysis of Fe(III). Hydrogen peroxide 10 mmol/L was dissolved in 100 mL of DDW with addition of 0.1 mL concentrated sulfuric acid to prevent disproportionation of hydrogen peroxide. Thiobarbituric acid (TBA) 1 g was dissolved in 100 mL of 50 mmol/L sodium hydroxide solution. Trichloroacetic acid (TCA) 5.6 g was dissolved in 100 mL of DDW. Antioxidant activity of tested samples against hydroxyl radicals generated by the Fenton reaction was compared with the antioxidant activity of gallic acid (GA) 10 mmol/L in the PBS. The solutions were pipetted into test tubes in duplicate. Solutions were mixed well and incubated in a water bath at 38 °C for 40 min. Then, 500 µL of TBA solution was added into each test tube, stoppered, mixed well, and incubated in a boiling water bath for 10 min. After boiling and uncorking, 500 µL of TCA solution was added into each test tube, mixed well, and cooled down in a beaker with tap water. The absorbance of the solutions was determined in a 1 cm cuvette at 532 nm against DDW using the spectrophotometer Shimadzu UV-1800. The percentage of inhibition (Hydroxyl [%]) was calculated. All determinations were performed at least four times.


**
*Ferric Reducing Ability of Plasma (FRAP) Assay*
**


The working FRAP reagents and other reagents were prepared according to the published method [77]. The only change was the increase in hydrochloric acid concentration to 50 mmoL·L^−1^ for dissolving 10 mmoL·L^−1^ TPTZ (2,4,6-tripyridyl-s-triazine) [74,75]. Aqueous solutions of iron (II) sulfate heptahydrate in the concentration range of 0–900 mmol L^−1^ were used for calibration at 600 nm (r = 0.9997). The assay was performed manually at room temperature. The solutions were pipetted into test tubes in duplicate. Solutions were mixed well, and the absorbance at 700 nm was recorded after 5 min with the spectrophotometer. A gallic acid solution with a concentration of 10 mmol L^−1^ was used for comparison. The FRAP of the samples (in µmol·L^−1^) was calculated by the formula: FRAP = (Sample − Blank) × 500/(Standard − Blank). All of the determinations of FRAP in the samples were performed at least four times. All the solutions were used on the day of preparation. The FRAP assay, which provides fast reproducible results, measures the ability of an antioxidant to reduce the ferric tripyridyltriazine (Fe^+3^-TPTZ) complex and produce the ferrous tripyridyltriazine (Fe^+2^-TPTZ) complex, which is blue in color. The activity was expressed in gallic acid equivalents (GAE) (FRAP [µmol·L^−1^]). All determinations were performed at least four times.


**Biological assay to evaluate phytotoxic effect of extracts**



**
*Model plants*
**


Model plant seeds were used for the evaluation of phytotoxic activity against two monocot species, *Triticum aestivum* L. and *Hordeum vulgare* L., and two dicot plant species, *Raphanus sativus* L. and *Sinapis alba* L. All seeds were obtained from the Breeding Research Center in Malý Šariš (Slovakia).


**
*Phytotoxic activity*
**


The following factors were considered in the experimental treatment: (i) test plants: (dicots: radish (*R. sativus* L.), white mustard (*Sinapis alba* L.) and monocots: winter wheat (*T. aestivum* L.) and barley (*Hordeum vulgare* L.); (ii) different extracts; and (iii) different extract concentrations. The extracts were dissolved in distilled water and diluted to prepare the desired concentrations. The total amount of solution was 100 mL. We used 50 mL, 25 mL, 10 mL, 1 mL and 1µL of extract in solution. Distilled water was used as control. Test seeds were surface-sterilized in 95% EtOH for 15 s and rinsed thrice in distilled water. Ten sterilized seeds were sown into each Petri dish (90 mm diameter) containing 5 layers of Whatman filter paper. In each Petri dish, 7 mL of extract of different concentrations or distilled water was added. Each treatment was prepared in triplicate. The Petri dishes were kept in a growth chamber (20 ± 1 °C, natural photoperiod, Sanyo, MLR-351H, Osaka, Japan). Germination was evaluated and the radicle length (cm) measured after 120 h.


**Statistical analysis**


General differences in the germination percentage and root length (i) between control and experiments, (ii) between experiments mutually and (iii) between different months of plant material collection were assessed using Student’s *T*-test with three levels of significance (*p* < 0.05; *p* < 0.01; *p* < 0.001). To depict observed differences, descriptive statistics were used. Simple linear regression analysis was used to assess possible correlations between germination percentages/root length and content of dry biomass. The T-test was also used to assess differences between extracts in dry biomass and total phenol, flavonoid and coumarin contents. Germination percentages and root lengths were correlated with the dry biomass and total phenol, flavonoid and coumarin contents using Spearman’s RS correlation test. Student’s T-test was performed in Excel, and all other statistical analyses were performed using PAST, Version 4.03. Analysis of variance (ANOVA, LSD method—least significant differences) was used to assess the differences in antioxidant activity of the extracts. Correlation coefficients were calculated according to Pearson. The coumarin content in the different harvest periods following the semi-quantitative analysis was compared using the Tukey test following one-way ANOVA calculation using GraphPad Prism software (version 8.4.3). A *p*-value < 0.05 was considered to be statistically significant.

## 5. Conclusions

The invasive *Heracleum mantegazzianum* can influence other plant species by allelopathy. Based on the results obtained through research focused on the inhibitory effect of an extract from the species *Heracleum mantegazzianum* on four native plant species (barley, summer wheat, white mustard and radish), we found differences in the effects of the extracts with respect to the collection period of the plant material from which each extract was produced, and also of the model plant on which the given extract was used. A significant effect on the germination and growth of seeds of barley (*Hordeum vulgare*) and summer wheat (*Triticum aestivum*) was shown by an extract made from leaves of *H. mantegazzianum* collected in the month of April. When investigating the effect of these extracts on white mustard (*Sinapsis alba*), we demonstrated the highest anti-germinative and phytotoxic activity for inhibiting seed germination and root growth after exposure of these seeds to extracts from leaves of *H. mantegazzianum* collected in May and June. The lowest percentages of germinated seeds and the smallest root lengths of radish (*Raphanus sativus*) seedlings were recorded with extracts made from hogweed collected in June and July. The correlation between chemical composition and phytotoxic activity was not evident. Our dataset detected no correlations between the determined furanocoumarins, phenols or flavonoids and the allelopathic effects of *H. mantegazzianum* extracts from different collection periods on seed germination or seedling root growth.

There is still space to enrich the investigation with different methods that can uncover the unseen metabolic processes that are responsible for the allelopathy between different plant species.

The allelochemicals identified can possibly act as locally produced compounds in new, potentially more ecological herbicides, but in specific doses and for specific target organisms.

## Figures and Tables

**Table 1 plants-13-01333-t001:** Dry matter, total phenols, total flavonoids, antioxidant activity against DPPH, hydroxyl and superoxide free radicals, and FRAP of aqueous extracts of *Heracleum mantegazzianum* leaves and seeds.

Parameter	Leaf Extract	Seed Extract	*p*
DM [g·L^−1^]	24.98 ± 0.08 a	11.65 ± 0.36 b	*p* < 0.001
Phenols [GAE mg·g^−1^ DM]	24.81 ± 0.68 a	15.92 ± 0.04 b	*p* < 0.001
Flavonoids [QE mg·g^−1^ DM]	7.82 ± 0.52 a	2.03 ± 0.05 b	*p* < 0.001
DPPH IC_50_ [µg·mL^−1^]	173.08 ± 5.84 a	386.64 ± 2.17 b	*p* < 0.001
Superoxide [%·g^−1^ DM]	2.83 ± 0.02 a	2.39 ± 0.12 b	*p* < 0.001
Hydroxyl [%·g^−1^ DM]	3.20 ± 0.48 b	7.35 ± 0.50 a	*p* = 0.002
FRAP [µmol·L^−1^·g^−1^ DM]	182.59 ± 18.01 b	465.68 ± 17.04 a	*p* < 0.001

Data represent the mean ± s.d. (standard deviation); a, b values indicate differences between extracts (ANOVA, LSD method); GAE = gallic acid equivalents; QE = quercetin equivalents; DM = dry matter.

**Table 2 plants-13-01333-t002:** Content of total phenols, total flavonoids, and individual phenolic compounds in *Heracleum mantegazzianum* water extracts prepared from leaves collected in April, May, June and July 2019.

HPLC R_t_ (min)	DAD λ_max_ (nm)	Compound	Content of Compound in Extract (µmol·g^−1^ DM) ^§^			
			April	May	June	July
13.8	205, 216, 259, 294	Protocatechuic acid *	0.78 ± 0.02 c	5.56 ± 0.11 b	6.54 ± 0.07 a	6.14 ± 0.36 a
14.2	217, 236, 324	Neochlorogenic acid *	0.66 ± 0.01 b	3.14 ± 0.03 a	0.05 ± 0.001 c	0.07 ± 0.001 c
17.5	217, 239, 325	Chlorogenic acid *	25.76 ± 0.62 a	13.72 ± 0.30 b	0.06 ± 0.001 c	0.17 ± 0.01 c
20.8	217, 238, 323	Caffeic acid *	2.22 ± 0.11 c	14.72 ± 0.07 a	2.28 ± 0.01 c	4.07 ± 0.05 b
22.5	203, 256, 354	Quercetin -6DH, -Hex **	12.66 ± 0.27 b	22.32 ± 0.30 a	1.59 ± 0.01 d	4.18 ± 0.04 c
23.1	217, 237, 325	Caffeic acid derivative	2.39 ± 0.04 a	1.63 ± 0.02 b	0.10 ± 0.004 d	0.18 ± 0.01 c
24.5	265, 345	Kaempferol glycoside (1)	1.03 ± 0.01 b	1.13 ± 0.01 a	0.25 ± 0.003 d	0.41 ± 0.02 c
24.9	265, 346	Kaempferol-3-O-glucosyl-7-O-rhamnoside **	22.94 ± 0.48 a	22.17 ± 0.36 a	3.36 ± 0.08 c	7.81 ± 0.07 b
25.1	203, 255, 348	Quercetin -6DH, -6DH **	5.58 ± 0.06 b	6.06 ± 0.09 a	1.90 ± 0.03 d	2.79 ± 0.12 c
25.4	203, 254, 354	Quercetin glycoside (1)	3.35 ± 0.06 b	5.36 ± 0.06 a	1.12 ± 0.01 d	1.96 ± 0.05 c
25.8	201, 257, 324	Coumarin derivative	9.93 ± 0.18 d	21.16 ± 0.15 a	20.31 ± 0.15 b	19.10 ± 0.31 c
26.6	227, 309	*p*-Coumaric acid *	0.28 ± 0.02 c	1.52 ± 0.02 a	0.21 ± 0.01 d	0.38 ± 0.01 b
28.0	264, 343	Kaempferol-3,7-di-O-rhamnoside **	20.33 ± 0.34 a	12.34 ± 0.15 b	3.09 ± 0.02 d	8.84 ± 0.08 c
28.7	203, 256, 354	Quercetin-3-O-glucoside *	1.54 ± 0.05 b	9.26 ± 0.07 a	0.45 ± 0.02 d	1.15 ± 0.02 c
33.0	265, 346	Kaempferol-3-O-glucoside *	1.40 ± 0.08 b	7.98 ± 0.09 a	0.48 ± 0.02 d	1.13 ± 0.01 c
33.4	203, 256, 348	Quercetin-3-O-rhamnoside *	0.31 ± 0.001 b	0.94 ± 0.01 a	0.21 ± 0.001 d	0.26 ± 0.001 c
33.7	203, 254, 354	Quercetin glycoside (2)	0.59 ± 0.02 b	1.72 ± 0.04 a	0.28 ± 0.01 c	0.59 ± 0.02 b
34.8	203, 254, 355	Quercetin glycoside (3)	2.47 ± 0.03 a	1.65 ± 0.01 b	0.41 ± 0.01 d	1.02 ± 0.01 c
38.0	265, 348	Kaempferol glycoside (2)	3.03 ± 0.04 a	1.36 ± 0.01 b	0.20 ± 0.004 d	0.63 ± 0.01 c
38.3	265, 342	Kaempferol-3-O-rhamnoside *	0.57 ± 0.03 c	5.50 ± 0.06 a	0.32 ± 0.02 d	0.90 ± 0.003 b
38.6	202, 255, 349	Quercetin glycoside (4)	0.94 ± 0.03 a	0.79 ± 0.02 b	0.25 ± 0.003 d	0.56 ± 0.01 c
40.3	264, 344	Kaempferol glycoside (3)	2.63 ± 0.03 a	1.06 ± 0.01 b	0.16 ± 0.004 d	0.71 ± 0.01 c
Total phenols content in extract (GAE mg·g^−1^ DM) ^§§^			36.18 ± 0.49 b	46.15 ± 0.40 a	27.35 ± 0.28 d	32.14 ± 0.47 c
Total flavonoids content in extract (QE mg·g^−1^ DM) ^§§^			18.55 ± 0.22 b	30.39 ± 0.39 a	11.09 ± 0.09 d	12.40 ± 0.20 c

Data are expressed as mean ± standard deviation. * corresponds to commercially available standard; ** corresponds to not commercially available standard (see Materials and Methods); ^§^ determined by HPLC-DAD; ^§§^ determined spectrophotometrically; -6DH: -6-deoxyhexose; -Hex: -hexose; GAE: gallic acid equivalents; QE: quercetin equivalents. Numbers in parentheses indicate unidentified glycosides of the corresponding flavonol. The results followed by different letters within a row indicate a significant difference between extracts (result of Tukey’s test).

**Table 3 plants-13-01333-t003:** HRMS and MS/MS data of detected compounds in the *Heracleum mantegazzianum* extract.

Peak	tR MS (min)	Measured (*m/z*)	Error (ppm)	Ionization	Formula	MS/MS Fragments	Proposed Metabolite	Levels	Ref.
1	3.34	163.0404	8.7	[M-H]+	C_9_H_6_O_3_	145.02980; 135.04554; 117.03527; 107.05095; 89.04071; 77.04083	Hydroxycoumarin isomer 1	3	Chemspider
2	3.46	163.0404	8.6	[M-H]+	C_9_H_6_O_3_	145.02939; 135.04582; 124.95849; 117.03521; 107.05205; 89.04073; 77.04047	Hydroxycoumarin isomer 1	3	Chemspider
3	5.72	187.0403	6.9	[M-H]+	C_11_H_6_O_3_	131.05089; 115.05592; 77.04094	angelicin	2	[40]
4	9.54	187.0392	0.3	[M-H]+	C_11_H_6_O_3_	131.0487; 115.0530	psoralen	2	[40,41,42]
5	9.91	217.0511	7.1	[M-H]+	C_12_H_8_O_4_	202.02720; 174.03240; 161.06106; 146.03751; 77.04081	bergapten	2	Chemspider
6	10.60	217.0511	7.1	[M-H]+	C_12_H_8_O_4_	202.02720; 174.03240; 161.06106; 146.03751; 77.04081	xanthotoxin	2	Chemspider
7	11.22	247.0611	4.1	[M-H]+	C_13_H_10_O_5_	232.03772; 217.01429; 189.01926	pimpinellin	2	Chemspider
8	11.61	247.0611	4.1	[M-H]+	C_13_H_10_O_5_	232.03772; 217.01429; 189.01926	isopimpinellin	1	STD

**Table 4 plants-13-01333-t004:** Content of individual coumarins (mean ±SD) in *Heracleum mantegazzianum* water extracts prepared from leaves collected in April, May, June and July 2019.

Compounds	April	May	June	July
hydroxycoumarin isomer **1** ^a^	22.89 ± 1.17 c	23.24 ± 1.03 c	8.56 ± 0.32 d	2.40 ± 0.25 e
hydroxycoumarin isomer **2** ^a^	0.88 ± 0.11 c	1.56 ± 0.11 c,d	0.29 ± 0.14 c,e	- f
angelicin ^b^	5.69 ± 0.12 c	5.81 ± 0.12 c	9.99 ± 0.31 d	11.93 ± 0.22 e
psoralen ^b^	2.09 ± 0.21 c	2.64 ± 0.08 c	3.77 ± 0.15 d	3.74 ± 0.10 c,d
bergapten ^b^	1.33 ± 0.16 c	1.47 ± 0.09 c	0.58 ± 0.09 d	1.52 ± 0.10 c
xanthotoxin ^b^	1.17 ± 0.11 c	1.25 ± 0.12 c	1.13 ± 0.10 c	0.88 ± 0.11 c
pimpinellin ^b^	0.66 ± 0.09 c	1.01 ± 0.10 d	0.52 ± 0.11 c	0.38 ± 0.11 c,e
isopimpinellin ^b^	0.25 ± 0.07 c	0.41 ± 0.12 c	0.77 ± 0.10 d	0.05 ± 0.02 e

Calculated using the calibration curve of coumarin “a” and isopimpinellin “b” as an upper index. Data are expressed as standard equivalent mg·g^−1^ DM. Different letters for means within a row indicate a significant difference between extracts (results of Tukey’s test). “-“, not detected.

**Table 5 plants-13-01333-t005:** Percentage of germinated seeds (%) after exposure to different doses of *Heracleum mantegazzianum* water extracts prepared from leaves collected in April, May, June and July 2019.

Extract Dose	Control	1 uL	1 mL	10 mL	25 mL	50 mL	100 mL
**Month**	**April**						
**Dry Mass**	**0 mg/mL**	**0 mg/mL**	**0 mg/mL**	**0.6 mg/mL**	**3.1 mg/mL**	**7.6 mg/mL**	**16.6 mg/mL**
*Raphanus sativus*	53.3 ± 20.8	83.3 ± 11.6	80 ± 10	53.33 ± 15.3	70 ± 20	10 ± 0 *	3.3 ± 5.8 *
*Sinapis alba*	93.3 ± 5.8	100 ± 0	96.7 ± 5.8	80 ± 26.5	43.3 ± 11.5 **	6.7 ± 5.8	3.3 ± 5.8 ***
*Triticum aestivum*	83.3 ± 15.3	80 ± 10	76.7 ± 5.8	96.7 ± 5.8	76.7 ± 11.5	35.0 ± 7.1	16.7 ± 15.3
*Hordeum vulgare*	76.7 ± 11.5	90 ± 10	90 ± 17.3	93.3 ± 5.8	100 ± 0	93.33 ± 5.8	6.7 ± 5.8 ***
**Month**	**May**						
**Dry Mass**	**0 mg/mL**	**0.2 mg/mL**	**0.2 mg/mL**	**1.7 mg/mL**	**5.2 mg/mL**	**10.4 mg/mL**	**20.4 mg/mL**
*Raphanus sativus*	70 ± 10	56.7 ± 11.5	80 ± 20	80 ± 0	40 ± 17.3	16.7 ± 5.8 ***	3.3 ± 5.8 ***
*Sinapis alba*	96.7 ± 5.8	96.7 ± 5.8	96.7 ± 5.8	83.3 ± 15.3	26.7 ± 20.8 **	10 ± 17.3 ***	-
*Triticum aestivum*	80 ± 10	90 ± 0	93.3 ± 5.8	83.3 ± 15.3	90 ± 10	83.3 ± 15.3	73.3 ± 5.8
*Hordeum vulgare*	73.3 ± 20.8	93.3 ± 11.5	96.7 ± 5.8	86.7 ± 5.8	96.7 ± 5.8	93.3 ± 11.5	76.7 ± 20.8
**Month**	**June**						
**Dry Mass**	**0 mg/mL**	**0.2 mg/mL**	**0.3 mg/mL**	**2.0 mg/mL**	**5.0 mg/mL**	**10.4 mg/mL**	**21.0 mg/mL**
*Raphanus sativus*	83.3 ± 5.8	73.3 ± 15.3	70 ± 0	36.7 ± 15.3	43.3 ± 25.2	13.3 ± 5.8 ***	-
*Sinapis alba*	93.3 ± 5.8	96.7 ± 5.8	63.3 ± 11.5	90 ± 10	50 ± 10	10 ± 10 ***	-
*Triticum aestivum*	90 ± 10	86.7 ± 5.8	86.7 ± 11.5	80 ± 10	86.7 ± 15.3	76.7 ± 25.2	63.3 ± 15.3
*Hordeum vulgare*	96.7 ± 5.8	96.7 ± 5.8	96.7 ± 5.8	96.7 ± 5.8	83.3 ± 20.8	80 ± 17.3	43.3 ± 28.9 *
**Month**	**July**						
**Dry Mass**	**0 mg/mL**	**0 mg/mL**	**0.2 mg/mL**	**2.1 mg/mL**	**5.2 mg/mL**	**9.7 mg/mL**	**19.9 mg/mL**
*Raphanus sativus*	63.3 ± 25.2	63.3 ± 11.5	7.7 ± 1.5	76.7 ± 15.3	53.3 ± 15.3	26.7 ± 11.5	-
*Sinapis alba*	90 ± 10	80 ± 10	80 ± 20	93.3 ± 5.8	70 ± 10	36.7 ± 5.8 ***	3.3 ± 5.8 ***
*Triticum aestivum*	90 ± 10	76.7 ± 5.8	86.7 ± 5.8	86.7 ± 11.5	93.3 ± 5.8	76.7 ± 11.5	43.3 ± 20.8 *
*Hordeum vulgare*	86.7 ± 5.8	96.7 ± 5.8	93.3 ± 11.5	90 ± 10	96.7 ± 5.8	93.3 ± 5.8	83.3 ± 11.5

Notes: Means ± SD of percentage of germinated seeds (%) with the results of Student *T*-test indicating significant differences between control and experiments in the percentage of germinated seeds: * *p* < 0.05, ** *p* < 0.01, *** *p* < 0.001; “-“, no germinated seeds.

**Table 6 plants-13-01333-t006:** Root length (cm) after exposure to different doses of *Heracleum mantegazzianum* water extracts prepared from plant material collected in April, May, June and July 2019.

Extract Dose	Control	1 μL	1 mL	10 mL	25 mL	50 mL	100 mL
**Month**	**April**						
**Dry Mass**	**0 mg/mL**	**0 mg/mL**	**0 mg/mL**	**0.6 mg/mL**	**3.1 mg/mL**	**7.6 mg/mL**	**16.6 mg/mL**
*Raphanus sativus*	1.69 ± 1.28	3.02 ± 0.87	2.05 ± 0.22	1.15 ± 0.39	1.07 ± 0.54	0.63 ± 0.12	0.1
*Sinapis alba*	3.39 ± 0.83	4.28 ± 1.22	3.93 ± 0.66	1.04 ± 0.30 **	0.86 ± 0.04 **	0.4 ± 0.14 *	0.1 ***
*Triticum aestivum*	2.60 ± 1.25	2.87 ± 0.83	3.37 ± 1.24	1.39 ± 0.16	1.04 ± 0.42	0.33 ± 0.01 **	0.1 ± 0 ***
*Hordeum vulgare*	3.54 ± 0.45	3.17 ± 0.02	3.44 ± 0.29	2.44 ± 1.10	213 ± 0.28	1.19 ± 0.25 ***	0.15 ± 0.07 ***
**Month**	**May**						
**Dry Mass**	**0 mg/mL**	**0.2 mg/mL**	**0.2 mg/mL**	**1.7 mg/mL**	**5.2 mg/mL**	**10.4 mg/mL**	**20.4 mg/mL**
*Raphanus sativus*	2.48 ± 0.36	2.09 ± 0.52	2.15 ± 0.07	1.57 ± 0.22	1.03 ± 0.46	0.57 ± 0.42 **	1 ± 0 **
*Sinapis alba*	4.76 ± 3.72	4.55 ± 0.88	4.87 ± 0.87	1.59 ± 0.21 ***	0.69 ± 0.27 ***	0.23 ***	-
*Triticum aestivum*	0.87 ± 0.32	1.27 ± 0.40	1.67 ± 0.97	1.23 ± 1.07	0.57 ± 0.12	0.43 ± 0.31 ***	0.3 ± 0.17 ***
*Hordeum vulgare*	3.57 ± 0.15	3.43 ± 0.49	3.11 ± 0.19	2.72 ± 0.76	3.03 ± 0.36	2.33 ± 0.75 *	1.59 ± 0.20 ***
**Month**	**June**						
**Dry Mass**	**0 mg/mL**	**0.2 mg/mL**	**0.3 mg/mL**	**2.0 mg/mL**	**5.0 mg/mL**	**10.4 mg/mL**	**21.0 mg/mL**
*Raphanus sativus*	2.81 ± 1.16	2.84 ± 1.24	2.92 ± 0.54	2.08 ± 0.66	0.58 ± 0.07 *	0.43 ± 0.21 *	-
*Sinapis alba*	4.84 ± 0.92	4.08 ± 1.42	5.0 ± 0.70 *	2.72 ± 0.95 *	1.17 ± 0.41 **	0.5 ± 0.28 **	-
*Triticum aestivum*	3.36 ± 0.34	3.65 ± 0.96	3.17 ± 1.18	2.31 ± 0.59	1.31 ± 0.11 ***	0.95 ± 0.11 ***	0.87 ± 0.12 ***
*Hordeum vulgare*	2.68 ± 0.57	3.89 ± 1.97	2.79 ± 0.19	2.16 ± 0.10	2.2 ± 0.23	1.83 ± 0.57	0.99 ± 0.52 *
**Month**	**July**						
**Dry Mass**	**0 mg/mL**	**0 mg/mL**	**0.2 mg/mL**	**2.1 mg/mL**	**5.2 mg/mL**	**9.7 mg/mL**	**19.9 mg/mL**
*Raphanus sativus*	2.41 ± 0.86	2.43 ± 0.16	2.26 ± 0.72	1.32 ± 0.23	0.86 ± 0.36 *	0.96 ± 0.29 *	-
*Sinapis alba*	3.32 ± 0.68	3.93 ± 1.59	3.42 ± 0.55	2.34 ± 1.24	1.72 ± 1.10	0.43 ± 0.23 **	0.7 **
*Triticum aestivum*	3.64 ± 0.67	3.27 ± 1.84	3.13 ± 0.72	1.91 ± 0.70*	1.80 ± 0.09 **	1.41 ± 0.29 **	0.41 ± 0.27 ***
*Hordeum vulgare*	3.06 ± 0.43	3.31 ± 0.26	3.87 ± 0.64	2.50 ± 0.68	2.11 ± 0.42	2.18 ± 0.33	0.75 ± 0.18

Notes: Means ± SD of with the results of Student *T*-test indicating significant differences between control and experiments in root length: * *p* < 0.05, ** *p* < 0.01, *** *p* < 0.001; “-”, no germinated seeds.

## Data Availability

Data are contained within the article.

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
