# Peer review of "Does the Invasive Heracleum mantegazzianum Influence Other Species by Allelopathy?"

_plants, 2024, doi:10.3390/plants13101333_

Round 1

Reviewer 1 Report

Comments and Suggestions for Authors

After reviewing this paper, the overall impression is that it is a mixture of a methodologically sound concept and unclear reasoning and interpretation of results. The topic is relevant, fits into the topic of SI, and the appropriate methodology has been applied. However, certain sections require complete rewriting to ensure coherence and alignment with the overall idea. Specifically, the Introduction needs to be both shortened and clarified. The research questions, hypothesis, goals of the study, and rationale for the chosen methodological approach should be explicitly defined. I would recommend a less educational and more scientific writing style in this section.

Although the results are presented clearly and concisely, their thorough explanation, discussion, and conclusion are missing. In the Discussion section, apart from comparing the results of other authors' studies, the authors made very few conclusions based on their own research. For instance, how do you interpret the correlations and differences in the antioxidant activity of extracts measured by different tests (Table 1)? Furthermore, what is the significance of the results presented in Table 2? Can you draw some conclusions from the total content of phenols and flavonoids (which are also phenols) and the individual components of the extracts, as determined based on the available standards? Are the results solely about the detected concentrations of selected compounds, or do they have broader implications? How do you interpret the higher seed germination after treatment with Heracleum extracts compared to the control (Tables 5 and 6)?

Please revise the explanation under Table 1 to include the type of statistical difference (marked with letters "a" and "b") based on the appropriate test. Additionally, ensure that all necessary marks are included where they are currently missing. The same applies to Table 2. Statistics related to the values shown in Table 4 and Figure 4 (line 248) are missing.

Comments on the Quality of English Language

The text contains numerous typos and requires thorough linguistic and stylistic editing.

Author Response

Dear Reviewer.

Thank you very much for your time and kind advice to improve our research article. Based on this we have provided following corrections and improvements.

Introduction was shortened, reformulated and the hypothesis with the precise explanation of the aim was added. The conclusion and discussion was also enriched with our own observations of the results and possible impact on other research.

Reply to your questions:

For instance, how do you interpret the correlations and differences in the antioxidant activity of extracts measured by different tests (Table 1)?

The all of the methods were used to confirm the total antioxidant activity of the extracts and each confirms the effect of a different spectrum of phenolic compounds

Furthermore, what is the significance of the results presented in Table 2?

Table 2 was corrected and added explanations

Can you draw some conclusions from the total content of phenols and flavonoids (which are also phenols) and the individual components of the extracts, as determined based on the available standards?

The sentence in the text was rewritten: Although there were significant differences between April, May, June and July extracts in the dry mass content, total phenols content, total flavonoids content, content of individual phenolic acids, flavonoids, and coumarins, no significant relationship between those quantitative parameters and the seed germination neither the seedling root growth was detected with our set of data. 

Are the results solely about the detected concentrations of selected compounds, or do they have broader implications?

In current research we have tested just selected concentrations. Mre study is required. We will continue in near future.

How do you interpret the higher seed germination after treatment with Heracleum extracts compared to the control (Tables 5 and 6)?

We suppose, that dry biomass in lower concentration can provide nutrient support for the seeds in comparison to pure water. That is why we can see higher germination rate expressed in %, however, there were no real differences confirmed though statistical evaluation within our set of data.

Please revise the explanation under Table 1 to include the type of statistical difference (marked with letters "a" and "b") based on the appropriate test. Additionally, ensure that all necessary marks are included where they are currently missing. The same applies to Table 2. Statistics related to the values shown in Table 4 and Figure 4 (line 248) are missing.

 added

Reviewer 2 Report

Comments and Suggestions for Authors

The topic covered in the paper is very interesting and has original aspects; however, many points need to be supplemented and improved. Below are my suggestions and requests for clarification.

Introduction:

- Better specify the term exotic invasive species: in the introduction it is written Plant species which spread from their native regions to different localities on different continents and are successful in new areas are called invasive species. This is not always true; there are exotic species that are successful and naturalize but do not become invasive. What are the causes that promote invasion? Certainly physiological, morphological, phenological and ecological causes intrinsic to the species, but also external causes. Does a disturbed and highly anthropized environment promote invasion? There are many literature sources to cite.

- There is a lack of overview of Heracleum species and their impact as invasives in different European countries. What are the most invasive species? What are the ecological, morphological, ecophysiological differences? Please consult the bibliography

-        Many biblographical sources could be cited, for example:

Pyšek P., Cock M. J. W., Nentwigw. & Ravn H. P. (eds.), 2007 - Ecology and Management of Giant Hog-weed (Heracleum mantegazzianum).CABI, Oxfordshire.

Sárka Jahodová, Sviatlana Trybush, Petr Pysek, Max Wade and Angela Karp - Invasive species of Heracleum in Europe: an insight into genetic relationships and invasion history. Diversity and Distributions, (Diversity Distrib.) (2007) 13, 99–114. DOI: 10.1111/j.1366-9516.2006.00305.x

KOUTIKA, L-S., RAINEY, H.J. , DASSONVILLE, N.- Impacts of Solidago Gigantea, Prunus Serotina, Heracleum Mantegazzianum And Fallopia Japonica Invasions on Ecosystems. APPLIED ECOLOGY AND ENVIRONMENTAL RESEARCH 9(1): 73-83.

Alm, T. Ethnobotany of Heracleum persicum Desf. ex Fisch., an invasive species in Norway, or how plant names, uses, and other traditions evolve. J Ethnobiology Ethnomedicine 9, 42 (2013). https://doi.org/10.1186/1746-4269-9-42

Koldasbayeva, D., Tregubova, P., Shadrin, D. et al. Large-scale forecasting of Heracleum sosnowskyi habitat suitability under the climate change on publicly available data. Sci Rep 12, 6128 (2022). https://doi.org/10.1038/s41598-022-09953-9

The Materials and Methods section should be brought forward after the introduction.

- How many samples of Heracleum were used? What is the collecting habitat (fallow, grassland, woodland edge..)? Is there no difference between the leaves of one Heracleum species and another? Between the seeds of one species and another? Perhaps it was worth making separate extracts of leaves/seeds of different Heracleum species to see if the concentration of phenols, free radicals, flavonoids was the same in all species or different. Probably within the same species, samples collected from populations in different geographic areas could give different results. Or not?

- However, the materials and methods section needs to be set out more clearly and coherently, perhaps a general outline of the approach to the work (to be presented as a figure) might help to better understand the method.

The results and discussion appear well presented, although since the materials and methods are not very clear, it is not clear if statistically the data are robust.

The conclusions are very concise. Better explain what the originality of the work consists of, what the most interesting results are, and what the practical spin-offs may be.

Although I am not a native English speaker, the text appears in good English.

Author Response

Dear Reviewer.

Thank you very much for your time and kind advices to improve our research article. Based on this we have provided following corrections and improvements.

Introduction was shortened, reformulated and the hypothesis with the precise explanation of the aim was added. The conclusion and discussion was also enriched with our own observations of the results and possible impact on other research.

The important thing which we changes is, that we corrected the Heracleum spp. into Heracleum mantagezzianum, which is the only identified invasive species in Slovakia  (refference number: 19)

Because the article is quite longwe tried to focus on the information closely realted to the experiments.

The template for the research article in journal Plants has order as i tis in the present manuscript, we are no table to change M&M after Introduction.

Round 2

Reviewer 2 Report

Comments and Suggestions for Authors

I have revised the corrected text. It seems to me that all the comments and corrections have been made. The methodology, results and discussion is now comprehensive and clear. I would say that we can proceed with publication.

Thank you, best regards